# Solving STODS—Surgical Temporary Ocular Discomfort Syndrome

**DOI:** 10.3390/diagnostics13050837

**Published:** 2023-02-22

**Authors:** Matthew T. Hirabayashi, Brad P. Barnett

**Affiliations:** 1Department of Ophthalmology, University of Missouri School of Medicine, 1 Hospital Dr, Columbia, MO 65212, USA; 2California LASIK & Eye, 1111 Exposition Blvd. Bldg. 200, Ste. 2000, Sacramento, CA 95815, USA

**Keywords:** LASIK, Keratopathy, dry eye, STODS, LALEX, SMILE

## Abstract

The term STODS (Surgical Temporary Ocular Discomfort Syndrome) has been coined to describe the ocular surface perturbations induced by surgery. As one of the most important refractive elements of the eye, Guided Ocular Surface and Lid Disease (GOLD) optimization is fundamental to success in achieving refractive outcomes and mitigating STODS. Effective GOLD optimization and the prevention/treatment of STODS requires an understanding of the molecular, cellular, and anatomic factors that influence ocular surface microenvironment and the associated perturbations induced by surgical intervention. By reviewing the current understanding of STODS etiologies, we will attempt to outline a rationale for a tailored GOLD optimization depending on the ocular surgical insult. With a bench-to-bedside approach, we will highlight clinical examples of effective GOLD perioperative optimization that can mitigate STODS’ deleterious effect on preoperative imaging and postoperative healing.

## 1. Introduction

Maintaining a healthy ocular microenvironment is requisite for tear-film stability and good vision [1]. In the wake of ocular surgery, all patients develop, in varying degrees, Surgical Temporary Ocular Discomfort Syndrome (STODS). To combat STODS, a molecular, cellular, and anatomic understanding of the ocular perturbations resulting from STODS is requisite. Armed exclusively with preservative-containing artificial tears, soap, and heat, numerous doctors and patients alike have found ocular surface microenvironment optimization to be an elusive goal.

STODS is a term popularized through the Refractive Surgery Alliance (https://www.refractivealliance.com/, (accessed on 6 December 2022)). Here, we use STODS to describe the temporary disturbance to the ocular surface following ocular surgeries involving incisions (manual or laser-assisted) to the cornea. The importance of STODS is its distinction from “dry eye disease”. The proposed draft for LASIK Patient Labeling Recommendations from the United States Food and Drug Administration also lists moderate dry eye symptoms as a relative contraindication for treatment, so confidently navigating preoperative ocular surface abnormality, optimization, and postoperative STODS will only become of even greater importance for refractive surgeons in years to come [2]. STODS is likely due to corneal nerve plexus transection and attenuated by other factors including up-regulation of inflammatory mediators. Corneal nerve fiber bundles are known to decrease significantly after procedures like LASIK but substantially return by one year postoperatively; hence the temporary nature of the condition (Figure 1) [3]. 

A stable, healthy tear film not only maximizes the quality and accuracy of preoperative measurements for surgical planning but also provides the greatest postoperative vision [4]. To achieve and maintain refractive targets, a careful coordination between optometrist, ophthalmologist, and patient is necessary. Effective co-management for the mitigation of STODS requires a proactive approach. This coordinated approach, or Guided Ocular Surface & Lid Disease (GOLD) optimization, is critical for achieving refractive targets and keeping patients 20/Happy. Surveillance for preoperative signs in the asymptomatic is of the utmost importance. Ensuring a stable tear film with non-invasive tear breakup time (NITBUT) and assessing glandular health with meibography is critical (Figure 2). As we learned from the Prospective Health Assessment of Cataract Patients’ Ocular Surface (PHACO), 80% of patients have clinically significant ocular surface disease prior to surgery but only 22% of these patients carry a diagnosis of dry eye disease (DED) [5]. While the importance of GOLD optimization and effective co-management for STODS mitigation is a growing topic of discussion, it remains a new topic with limited literature on what approaches are most efficacious or efficient. 

Although it is always important to consider optimizing the ocular surface for all patients to improve their vision, it is of critical importance before cataract surgery. Cataract surgery is considered a form of refractive surgery with patients expecting excellent postoperative results with clear and stable vision regardless of the type of intraocular lens (IOL). In this paper, we will discuss how health of the ocular surface is assessed, what molecular changes refractive surgery induces, and how targeted therapies aim to enhance GOLD optimization and prevent, as well as treat, STODS should it occur.

This review, with a few case studies, attempts to take a bench-to-bedside approach to solve and effectively combat STODS and attenuate concomitant disruption of ocular surgery to provide the greatest chance for symptom minimization.

## 2. The Importance of Preoperative Optimization

First, patients should be counseled that an optimized ocular surface provides the greatest probably of the most accurate preoperative measurements possible, and thus, the most accurate outcome possible. This applies to keratometry, corneal tomography or topography, and biometry as well as current manifest refraction. Without a regular and stable ocular surface before surgery, patients should understand they might not be able to realize their full visual potential. Rapid tear breakup time (TBUT), punctate epithelial erosions, or low tear lakes with desiccation affect the reliability, reproducibility, and accuracy of the preoperative measurements (Figure 3) [6]. This therefore increases the possibility of a surgeon making an inappropriate IOL recommendation or selecting an incorrect IOL power leading to suboptimal visual outcomes for patients. A suboptimal ocular surface can have a profound impact on topography measurement [7]. 

Multifocal and extended depth of focus (EDOF) presbyopia-correcting IOLs provide the greatest change for spectacle independence following cataract surgery but are exquisitely sensitive to ocular surface disease with more variability and dissatisfaction with vision than monofocal patients (Figure 4) [8,9,10]. These patients should be specifically counseled that GOLD optimization will remain paramount after surgery for the best possible outcome.

## 3. Key Elements to Preoperative Evaluation

Whether patient is an outside referral or established in the practice, the surgeon must evaluation their corneal health preoperatively. There are components to the ocular surface exam with some clues evident when entering the room. Excessive or frequent blinking, skin manifestations of rosacea, blepharitis, incomplete blink, lagophthalmos, or eye rubbing are often evident from across the room. Elucidating details regarding contact lens wear, ocular comfort, or symptoms of dry eye or blepharitis also help in guiding both the exam and final IOL recommendation.

Classic clues for ocular surface abnormality include anterior blepharitis, meibomian gland dysfunction, punctate corneal staining, anterior basement membrane corneal dystrophy, or Salzman’s nodules, many of which are common in the cataractous population [12,13]. For a more objective analysis of the ocular surface, corneal tomography/topography shows fluctuating and significant irregularity [7]. A traditional way to assess the corneal tear film is tear breakup time (TBUT). This is an “invasive” assessment that involves placing fluorescein in the tear film and timing its evaporation, which is considered normal if >10 s [14]. 

New methods such as NITBUT have been gaining popularity and typically involve video topography (Figure 2) [15]. Examples include the CA-800 (Topcon, Tokyo, Japan), TearCheck (ESW Vision, Houdan, France), and Keratograph 5M (Oculus, Wetzlar, Germany). Results are comparable to traditional TBUT but may be more repeatable and reliable since they are a noninvasive test [16]. Importantly, it can be incorporated with a dropless preoperative evaluation that will not interfere with topography. As many groups use an anesthetic/fluorescein solution for applanation tonometry and TBUT, assessment of corneal sensation with an esthesiometer is precluded. As corneal sensation may play a central role in STODS, it is important to assess corneal sensation both pre- and postoperatively. For this reason, devices such as iCare (Vantaa, Finland), that do not require anesthetic for use, may be ideal in the refractive surgery setting.

## 4. Molecular Changes in Ocular Surface Abnormality and during Refractive Surgery

Striving for GOLD optimization and the prevention/treatment of STODS requires an understanding of the molecular factors that influence ocular surface changes during surgery. Several mechanisms have been proposed and studied. Cataract surgery alone leads to ocular surface changes and dry eye syndrome through several mechanisms that disrupt tear film stability [17]. Corneal nerve destruction (Figure 1) during wound creation, triggering the inflammatory cycle, goblet cell loss, and meibomian gland dysfunction have all been reported after cataract surgery [18,19]. Ocular surface inflammation appears to play a dominate role over tear secretion [17]. Longer operative times, light or heat from the microscope, use of a lid speculum, and the severity of intraocular inflammation can all impact the postoperative ocular surface [20]. Misuse of postoperative drops is also among the major contributors to STODS [21].

One of the best studied theories involves a cycle of ocular surface inflammation comprised of both soluble and cellular mediators [22]. For example, patients with and without Sjögren’s syndrome appear to have identical T-cell activation and infiltration with upregulation of CD3, CD4, CD8, CD11a, and HLA-DR, the latter two markers specific to lymphocyte activation [23]. Additionally, increased inflammatory cytokines such as interleukin 1 (IL-1) and upregulation of matrix metalloproteinases (MMPs) have been demonstrated in the tear film of patients with dry eye symptoms and ocular surface diseases [24]. Other responses to ocular surface stress include hyperosmolarity and increase in MMPs mediated by intracellular pathways including mitogen-activated protein (MAP) kinase and inflammatory cytokines. This results in a cycle where hyperosmolarity then induces inflammation of limbal epithelial cells by further upregulating inflammatory cytokines [25].

The degree to which patients have these various perturbations at baseline and their susceptibility to such perturbations has underlying genetic causes [3]. Rosacea and elevated levels of MMPs are an area ripe for exploring genetic underpinnings that can contribute to STODS. It has been reported that 80% of persons with rosacea have concurrent MGD. Though poorly understood, early research suggests gene-gene and gene-environment interactions are central to rosacea’s developments. Whether through inherited genes or through epigenetic modifications that occur through environmental influence, understanding how genes impact STODS is critical [25].

It does appear that certain ocular insults result in chronic ocular surface disease. So instead of STODS, a patient develops Surgical Chronic Ocular Discomfort Syndrome or SCODS. The genetic risk factors as well as epigenetic modifications that occur in the setting of ocular surgery are areas requiring further research. For example, currently we have no evidence to support the idea that epigenetic modifications occurring in longstanding STODS may contribute to the development of SCODS. Alternatively, the insult may cause purely neurogenic and inflammatory changes that are better blunted by certain genetic predispositions in lieu of epigenetic modifications.

Additionally, dysregulation of the balance between proteases and protease-inhibitors has been observed in ocular surface disease. These include MMPs but also cathepsins and plasminogen activators (and their relevant inhibitors) [26]. MMPs, serine proteases, and cysteine proteases are all shown to be upregulated in ocular surface disease and all play a role in protease-activated receptor (PAR) inflammatory signaling [26].

The concept of the eye biome has increased in popularity recently with evidence for a distinct microbiome in those with dry eye disease compared to healthy individuals [27]. As expected, those with a blepharitis component have increased prevalence of *Streptophyta*, *Corynebacterium*, and *Enhydrobacter* [28]. Meibomian gland dysfunction has been associated with increased bacteria on the ocular surface and an increase in bacterial load has been associated with decreased goblet cell density [29,30]. In this way, alterations in the ocular microbiome have major effects on tear film stability.

Since the most optimal therapy would involve tailoring the treatment to the pathophysiology, certain surgical techniques induce unique changes. Following Photorefractive Keratectomy (PRK), new neurites emerge from the severed nerve endings into the epithelial-stroma interface as early as the first week after surgery [31]. There is around 85–95% sensitivity recovery after PRK by 3 months that is directly related to the intensity of the laser application [32,33,34,35]. Laser In-Situ Keratomileusis (LASIK) on the other hand severs both stromal and sub-basal nerves during flap creation with direct ablation of the stromal nerve plexus [36,37]. There is typically under 10% of the sub-basal nerves remaining after LASIK with evidence for both continued regression after surgery since the nerves are unable to connect with the flap leading to significantly reduced nerve density up to 5 years postoperatively (Figure 1) [3,38,39,40,41,42,43]. Laser-Assisted Lenticule Extraction (LALEX) shows superior postoperative corneal sensitivity compared to LASIK, likely because the nerves outside the lenticule area remain untouched [37,44]. Femtosecond lasers used for cataract surgery (e.g., capsulotomies) have also been associated with a dose-dependent induction of cell-induced inflammation [45]. The use of the femtosecond laser in LASIK surgery does not appear to alter corneal sensitivity or dry eye outcomes compared to LASIK alone. In fact, femtosecond flaps may have superior postoperative tear film stability compared to mechanically created flaps [46,47,48]. Cell-mediated inflammation still plays a major role in ocular surface disease following PRK, LASIK, and LALEX [21].

The emergence of Microinvasive Glaucoma Surgeries (MIGS) and the “MIGS Revolution” has allowed for the intervention of glaucoma earlier in the disease course and is often paired with intraocular surgery often with the goal to reduce the medication burden and lessen ocular surface side effects of topical glaucoma therapy [18]. There is currently limited literature concerning unique variations of inflammation when cataract surgery is performed with MIGS but there is no clear evidence that MIGS increase inflammation postoperatively when performed correctly. Although it is important to consider glaucomatous eyes can have a unique profile of inflammatory cytokines [49].

Thus, it is clear that ocular surface and dry eye disease is a complex entity with delicate interplay between inflammatory mediators, tear film integrity, and the native microbiome and these all must be considered when attempting to treat patient symptoms.

## 5. Methods for Preop Ocular Surface Optimization

Lifestyle changes, although challenging to introduce, are important to consider for GOLD optimization. Discussing the impact of environmental factors with patients such as overhead fans, placement of vents, and any other source of blowing air that would contribute to tear desiccation is critical. For people with incomplete lid closure and/or Continue Positive Airway Pressure (CPAP) use, nighttime ointment or occlusive mask during sleep is critical. Prolonged reading or binge-watching media will also exacerbate symptoms of dry eye [50]. It may be best at these early stages of treatment to remind patients that the surgery discussion must wait until the ocular surface improves and accurate measurements are possible.

The most pervasive first-line treatment for dry eye symptoms are artificial tears or ocular lubricants which generally result in a ~25% improvement of symptoms [51]. The issue is, artificial tears, although soothing even when preservative free, are not wholly benign and do not effectively restore ocular surface perturbations. Other than the immediate symptomatic relief they provide, they also have a “dilution” effect of the inflammatory mediators. Starting newly diagnosed patients on artificial tears QID and placing collagen punctal plugs such as the DuraPlug (Katena, Denville, NJ, USA) that will dissolve after 3–4 months are reasonable first-line treatment options for mild disease before and after surgery. While there is some evidence for the efficacy of lid scrubs and warm compresses, these methods do not appear to alter lipid layer thickness or tear interferometry [52]. Additionally, perhaps it is not the best initial step in therapy to apply soaps that may further disrupt the biome, mechanical scrubbing, or heat to tissue that is already inflamed.

For primarily lid margin disease or blepharitis, especially with *Demodex*, hypochlorous acid-containing lid scrubs such as Avenova^®^ (NovaBay Pharmaceuticals, Emeryville, CA, USA) or OCuSOFT (OCuSOFT Inc., Richmond, TX, USA) can provide effective relief [53]. Lotilaner Ophthalmic Solution has also demonstrated efficacy specifically for *Demodex* treatment [54]. Warm compresses are particularly useful for blepharitis because since the meibomian gland is a holocrine gland, it is similar in concept to skin exfoliation in that expression can regenerate atrophic glands [55]. Devices such as LipiFlow^®^ (Johnson & Johnson, New Brunswick, NJ, USA) or TearCare^®^ system (Sight Sciences, Menlo Park, CA, USA) are more technologically sophisticated than warm compresses but use the same principle of heat to express meibomian glands with the goal of resetting the microenvironment [56,57]. A similar method for addressing meibomian gland disease is Intense Pulsed Light (IPL) therapy. Light is converted to heat that ablates vessels and restricts the inflammatory mediators to the gland structures and has proven effective for abating symptoms of dry eye [58]. The E-EYE IRPL^®^ (ESWIN, Houdan, France) was the first registered and medically certified IPL for the DED & MGD in 50+ countries outside of the U.S. Now available in the U.S. along with Optilight^®^ (Lumenis, Yokneam, Israel), we have IPL devices with proven efficacious protocols for optimizing ocular surface disease.

To address more of the root causes of ocular surface disease, topical anti-inflammatories have been growing in popularity. Lifitegrast ophthalmic solution (Xiidra^®^, Novartis AG Pharma, Basel, Switzerland) addresses this by binding to lymphocyte function-associated antigen-1 (LFA-1) to prevent the inflammatory cascade with good efficacy in treating symptoms [59]. A T-Cell suppressant regimen such as Xiidra^®^ BID, a topical steroid such as Eysuvis^®^ (Alcon, Fort Worth, TX, USA) QDAY prior to surgery are reasonable approaches to address the inflammatory component. Similarly, topical cyclosporine A including cyclosporine ophthalmic solution 0.09% (Cequa, Sun, Princeton, NJ, USA) and cyclosporine ophthalmic solution 0.05% (Restasis^®^, AbbVie, Chicago, IL, USA) as well as new generic 0.05% cyclopsporine formulations, inhibit T-cell activation and production of inflammatory cytokines by inhibition of calcineurin [60,61]. 

A regimen such as two months of Cequa^®^ BID and Eysuvis^®^ (Alcon, Fort Worth, TX, USA) QDAY with potential erythromycin ointment nightly for MMP-9 inhibition is another reasonable way to approach GOLD optimization from the inflammatory perspective before surgery. The use of topical corticosteroids has been proven effective for the signs and symptoms of ocular surface disease [62]. Strategic use of these inflammatory mediators must be considered with other therapies. For example, full punctual occlusion with lifitegrast results in new tears diluting the lifitegrast until it dissociates and makes the LFA-1 accessible to enable T-Cell ocular surface invasion. 

While tetracyclines do reduce bacterial flora, they also have been shown to inhibit lipase activity, MMP-9 levels, and inflammatory cytokines [63]. A short course of doxycycline would also be a reasonable approach before ocular surgery. Lastly, while autologous serum tears derived from blood samples theoretically have the benefit of containing other biochemical factors to mimic an individual’s tear film more closely, they are difficult to obtain and evidence for efficacy is limited in the long term [64]. 

An alternative to the previously mentioned autologous serum tears is a hypoosmolar ophthalmic solution with a “patented formula” of human placental-derived biomaterials from amnionic fluid such as Regener-Eyes^®^ (Regener-Eyes^®^, Palm Harbor, FL, USA) which is typically dosed 1–4 times per day for symptoms of dry eye [65]. An immune-privileged site, amnionic fluid contains several factors to promote cell growth and regeneration of collagen [66]. Cell-free amnionic fluid derivatives have also proven effective in not only cases of corneal wound healing but also for symptoms of keratoconjunctivitis sicca [67]. Another approach is to place a few drops of these solutions into an ocular surface drug depot system such as Hyper-CL™ (EyeYon Medical, Ness Ziona, Israel). This theoretically protects the ocular surface from the mechanical action of blinking and increases the ocular surface contact time with the solution. Nevertheless, for applications that involve optimizing corneal topography/tomography, the presence of a contact lens that may induce astigmatism may not be the ideal approach.

To address reducing MMPs further, low-dose oral options such as doxycycline 40 mg (Oracea^®^, Galderma, Lausanne, Switzerland) or doxycycline hyclate 20mg (Periostat^®^, Galderma, Lausanne, Switzerland) provide improvement of meibomian gland dysfunction without disrupting the microbiome of the eye [68]. Erythromycin or even postoperative antibiotic therapy may disrupt this biome further so these options with minimal antibiotic properties or favoring intracameral antibiotics over a course of topicals may prevent STODS and help optimize the ocular surface. 

Similarly, the “pro-biotic” concept is relatively new to ophthalmology but treatments such as HydroEye^®^ (Spring, TX, USA) provide a proprietary blend of Gamma-linolenic acid (GLA), omega 3 fatty acids (EPA and DHA), antioxidants, and vitamins to support the tear film. The ingredients have demonstrated success and are another reasonable option both before surgery for GOLD optimization and postoperatively for the prevention of STODS [69]. Another benefit of oral preparations is patient acceptance. They are comfortable with taking a natural option long-term and appreciate that the anti-inflammatory effects can also have systemic benefits. These are also excellent options as therapy by an alternate route may reduce the future need to add more topical medications.

Another “minimally invasive” option is varenicline nasal spray (Tyryva™, Oyster Point Pharma, Princeton, NJ, USA) which has shown excellent efficacy and speed to clinical effect [70]. Neuroinflammation of the corneal nerves, trigeminal ganglion, and trigeminal brainstem complex have been associated with dry eye changes so this provides a promising new approach especially since it is not another topical therapy [71]. A month or two of this therapy is reasonable before surgery and can easily be continued after as well.

A step past the amniotic fluid-derived therapies is amnionic membrane grafts (AMG) which are composed of epithelial cells, their basement membrane, and a matrix of connective tissue all bathed in amnionic fluid containing numerous anti-inflammatory, immunomodulatory cytokines, and growth factors [72,73]. Cryopreserved AMG has shown promise in regeneration of corneal nerve and acceleration of ocular surface recovery in cases of dry eye [74]. Cryopreserved AMG can also be placed under a generic bandage contact lens or even a Kontur Contact Lens (Kontur, Hercules, CA, USA) for easy application (Figure 5).

Unlike dehydration, lyophilization is a gentler process that is less likely to denature proteins and thus, once rehydrated, results in more therapeutic potential [75]. Lyophilized autologous serum tears have demonstrated equivalent efficacy to fresh samples [76]. Lyophilized AMGs have been successfully used in pterygium surgery and are likely reasonable for other ocular surface disease applications as well [77]. Either lyophilized or cryopreserved AMGs are reasonable intermediate steps under contact lenses but additional studies, especially molecular analysis, will be necessary to tease out details if one is superior to the other.

Neox^®^ Flo (BioTissue, Miami, FL, USA) is another tissue product containing both amnionic membrane and umbilical cord (AM/UC) that has already demonstrated efficacy to assist in wound healing [78]. Demonstrated ophthalmic applications are currently limited but it will likely provide a promising option when used under a Kontur or bandage contact lens. Experiments are underway to compare NEOX^®^ FLO to Prokera^®^ to determine the utility for optimizing the ocular surface prior to biometry. Both this product as well as a new lyophilized amniotic membrane graft, Xcellereyes^®^ (Oculus Biologics, Willowbrook IL, USA) may be superior to Prokera^®^ for preoperative optimization as the retaining ring of the Prokera^®^ can ride up onto cornea, inducing astigmatism (Figure 5). The Prokera^®^ line of products can also leave a residue on the ocular surface when first removed. This paradoxically worsens the NITBUT and accordingly the quality of topography (Figure 5). Logistically, due to this residue and the potential astigmatism induced by the retaining ring, same day Prokera^®^ removal followed by biometry is non-ideal. In our current protocol, we stagger topography and biometry three days after Prokera^®^ removal to mitigate these potential confounders (Figure 6). Dehydrated AMGs under contact lenses do not leave a similar residue and the use of a large diameter contact lens such as a Kontur may be less likely to induce astigmatism and effectively retain therapeutic factors on the ocular surface (Figure 7). Experiments are underway to better understand if substitution of a lyophilized product under a contact lens may provide the superior therapeutic benefit of Prokera^®^ compared to dehydrated AMG without the residue deposition and astigmatism induction by the Prokera^®^.

## 6. Intraoperative Management

As discussed earlier, preserving the ocular microbiome may play a major role in managing ocular surface disease. Intraocular methods such as the newly described “Shimada Technique” employs 0.25% Povidone–Iodine washes during cataract surgery to incredible antimicrobial effect [79]. This technique, or intracameral antibiotics at the conclusion of surgery, may remove the need for postoperative antibiotics that continue to disrupt the ocular microbiome for the week following surgery [80]. Faster return to normal flora may equate to faster return to a healthy ocular surface. Use of dispersive viscoelastic such as OcuCoat^®^ (Bausch + Lomb, Laval, QC, Canada) on the ocular surface instead of a balanced saline solution may also prevent intraoperative drying and epithelial damage.

For laser-based surgeries, several of the previously mentioned considerations can further guide laser selection. Eyes with significant ocular surface disease may experience less STODS with PRK or LALEX over LASIK, for example. Additionally, femtosecond-assisted cataract surgery should be kept to the minimal necessary power. This concept of “low-energy” LALEX has gained popularity recently as technology continues to advance [81]. While LASIK flap thickness or hinge position have not demonstrated an effect on ocular surface symptoms, there is evidence that smaller flap diameters do create less nerve transection, and accordingly less ocular surface disease [82,83]. Thus, the ratio between the corneal and flap diameter is another consideration surgeons can use to minimize the risk of STODS. Another consideration is the demonstrated symptom profile of femtosecond created flaps compared to mechanically created flaps [47,48]. 

## 7. Postoperative Management of STODS

Any laser or cataract surgery can result in postoperative disruption to the tear film and cause Surgical Temporary Ocular Discomfort Syndrome (STODS) (Figure 8). Staying ahead of the inflammatory cascade using the methods mentioned above is likely the best way to prevent STODS, but there are ways to manage it should it occur. When it does occur, it is also essential to carefully pair the appropriate therapeutic target with the underlying pathophysiology of the cause.

Many of the previously mentioned therapies can be continued into the postoperative period, especially those that have routes other than application onto the ocular surface such as HydroEye^®^ or Tyryva™. As mentioned before, avoiding postoperative antibiotics may lead to faster restoration of the native ocular microbiome. Similarly, avoiding postoperative corticosteroid drops likely benefit the ocular surface by reducing exposure to preservatives. Products like the dexamethasone eluting punctal plug Dextenza^®^ (Ocular Therapeutix, Inc., Bedford, MA, USA) or subconjunctival triamcinolone can spare the ocular surface after surgery and provide effective control of intraocular inflammation [84,85]. Of the therapies that can or should be continued, cylcosporine 0.05% has been shown to improve visual outcomes after multifocal IOL implantation [9]. 

As referenced earlier, laser-based surgeries, including femtosecond-assisted cataract surgery, exert unique changes on the ocular surface (Table 1). In addition to cell-mediated inflammation, the destruction of corneal nerves and incited inflammation from Femto, LASIK, PRK laser-based surgeries implies these patients may not only benefit from medications with cellular anti-inflammatory mechanisms, but also therapies that assist with corneal nerve regeneration. Examples of this would include the amnion-derived bio-tissues discussed earlier such as NEOX^®^ FLO (Table 2).

## 8. Conclusions

Recognizing the signs of ocular surface disease is paramount in any surgical workup. This should be followed by an earnest discussion about the importance of GOLD Optimization for obtaining optimal measurements and thus optimal outcomes. Part of this discussion with patients is ensuring they understand the major role they play in their own outcome. Patient compliance is a known issue with dry eye treatment and the degree to which they comply with the treatment options surgeons suggest will directly correlate to their outcome [86]. Investing in their eyes, and their vision, is a lifetime commitment that starts before ocular surgery. Many new treatments for STODS and general dry eye symptoms are constantly developed so this remains an active and innovative area of research. Patents already exist for ways to regulate blood flow to restore a more normal oxygen status to dysfunctional meibomian glands [87]. 

Understanding that there’s no definitive cure for dry eye and the patient will never be able to abandon caring for their ocular surface is perhaps the best way to ensure happy patients and happy surgeons. In particular we must understand that although STODS may be temporary, if left untreated it can create a chronic inflammation or Surgical Chronic Ocular Disease Syndrome (SCODS). The effective refractive surgeon will identify the various intrinsic factors such as genetic propensity, ocular surface disease prior to surgical intervention, and various anatomical factors that may increase the risk of STODS to SCODS conversion. They will not be flat-footed and reactionary, but instead, proactive in mitigating STODS at the gate and ensuring it never becomes SCODS.

## Figures and Tables

**Figure 1 diagnostics-13-00837-f001:**
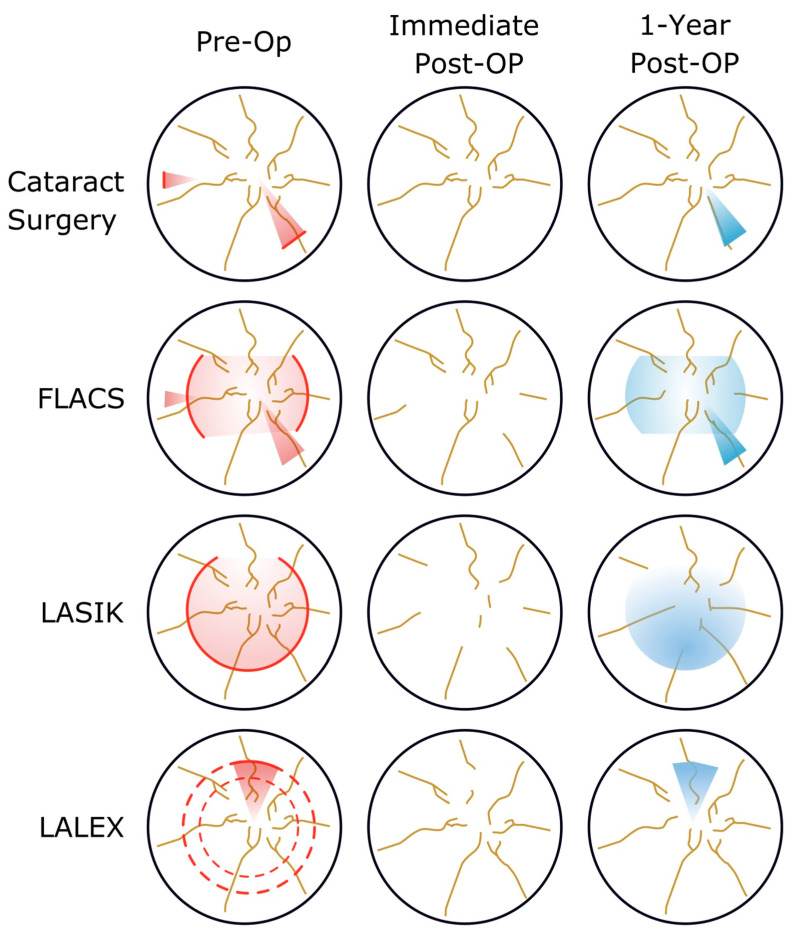
Neurotrophic Etiologies for Surgical Temporary Ocular Discomfort Syndrome (STODS) with a variety of corneal surgical interventions. Transection of nerve occurs in a varying amount with the variety of corneal surgeries as compared to cataract surgery without femtosecond arcs. When femtosecond arcs are employed, the potential for greater nerve transection and thus STODS develops. Similarly, LASIK transects more corneal nerves than LASIK; therefore one would expect greater STODS in the former surgery. Fortunately, by one year, all corneal surgeries are followed by robust nerve regrowth to nearly baseline levels.

**Figure 2 diagnostics-13-00837-f002:**
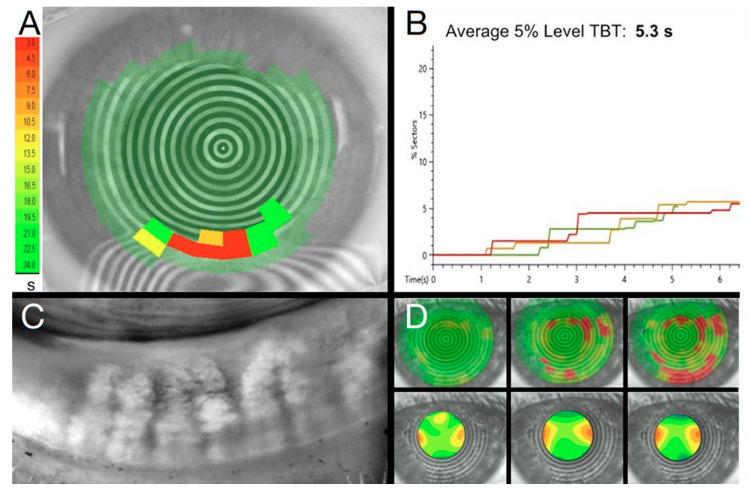
Non-invasive preoperative assessment with CA-800 (Topcon) Tear Film Analyzer. (**A**) Placido disk non-invasive tear breakup time (NITBUT) provides an objective measure of tear film stability (**B**). Through use of meibography (**C**) to demonstrate to patients contributing factors to tear film instability NITBUT with Zernike analysis of the resulting astigmatism, spherical aberration, coma, and higher order aberrations (**D**) to convey to patients the visual impact of tear film instability, clinicians can help patients better understand the impact of ocular surface disease even prior to surgery. Moreover, this analysis helps gauge IOL suitability and the likelihood of quality preoperative topography. A NITBUT of 1 s, for example, will undeniably result in poor topography as a Pentacam is captured over 2 s.

**Figure 3 diagnostics-13-00837-f003:**
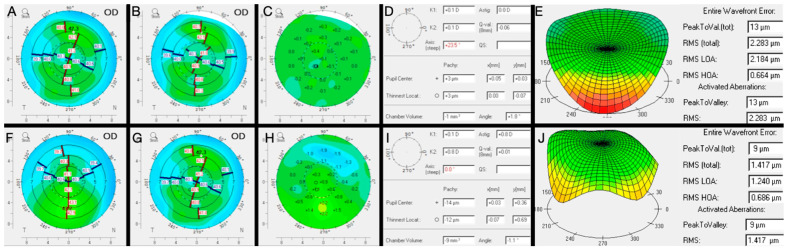
The impact of tear film stability on corneal topography. Sequential same day (**A**,**B**) Scheimpflug image in the setting of unstable tear film demonstrating a difference (**C**) of 23.5 diopters on the axis of astigmatism (**D**). In addition to varying magnitude and axis of astigmatism, tear film instability also results in increased wavefront error (**E**). Repeat topography after GOLD optimization reveals a more regular astigmatism with less same day variability (**F**,**G**). The best fit topography prior to GOLD optimization (**A**) was compared to (**F**) which revealed a 0.8D increase in magnitude in astigmatism (**H**,**I**) as well as a decrease in wavefront error (**J**).

**Figure 4 diagnostics-13-00837-f004:**
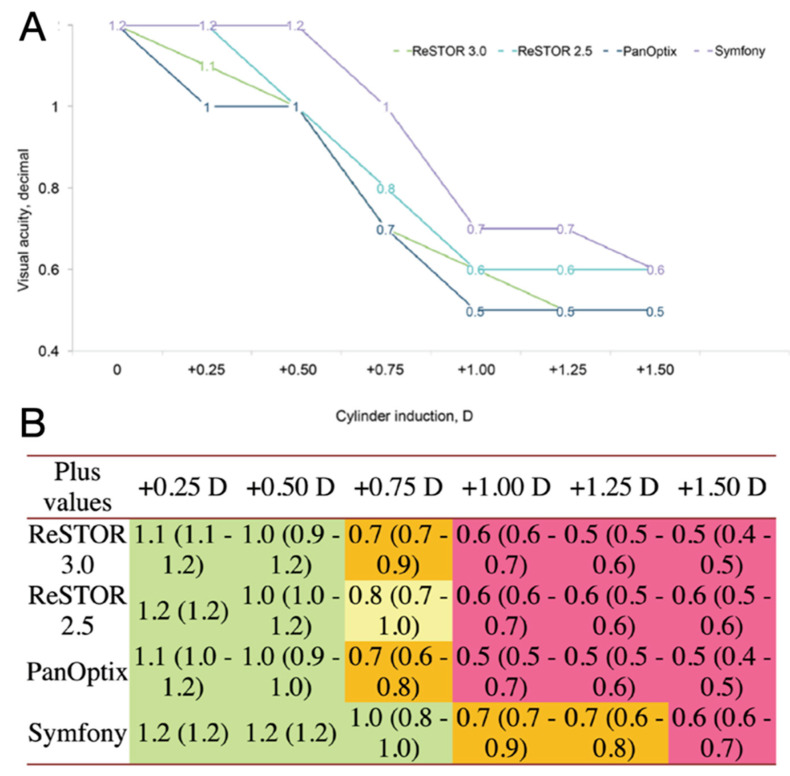
Mean visual acuity with four multifocal IOLs after the induction of different values of positive cylinder. Mean visual acuity (**A**) and patient satisfaction scores (**B**) with four multifocal IOLs after the induction of different values of positive cylinder (green 1/4 very satisfied; yellow 1/4 moderately satisfied; orange 1/4 not satisfied; red 1/4 not at all satisfied). Values are reported as median, with range in brackets [11].

**Figure 5 diagnostics-13-00837-f005:**
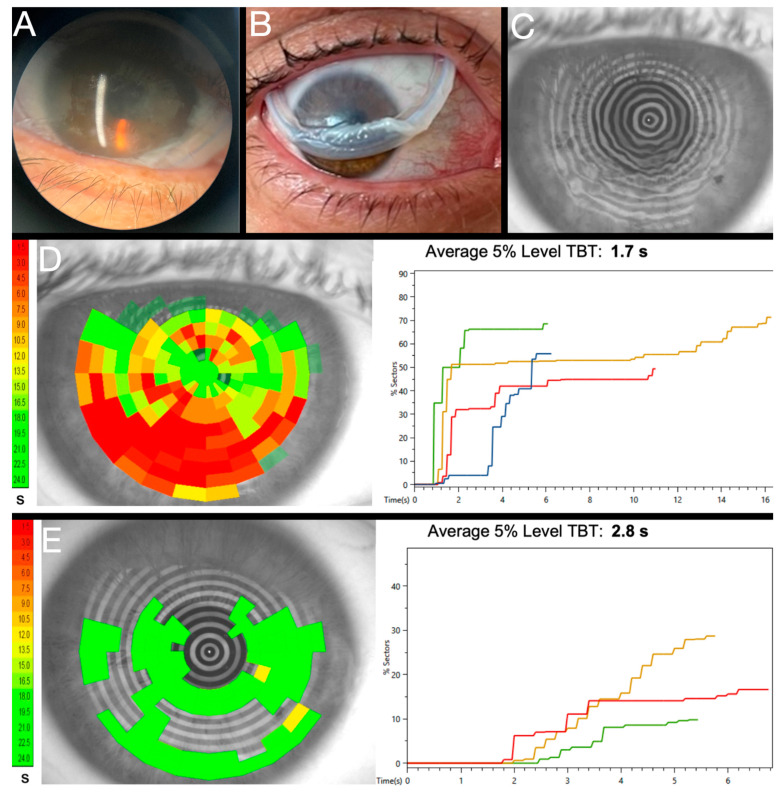
Prokera for preoperative STODS mitigation (**A**) demonstrates a view at the slit-lamp of standard Prokera placement (Note: even in best case scenario retaining ring presses on cornea). (**B**) Prokera has ridden up on cornea therapy inducing artifactual astigmatism. When immediately removed, Prokera leaves a film on the cornea (**C**) that interferes with the mires and paradoxically results in a worsened NITBUT (**D**) as compared to prior to insertion (**E**).

**Figure 6 diagnostics-13-00837-f006:**
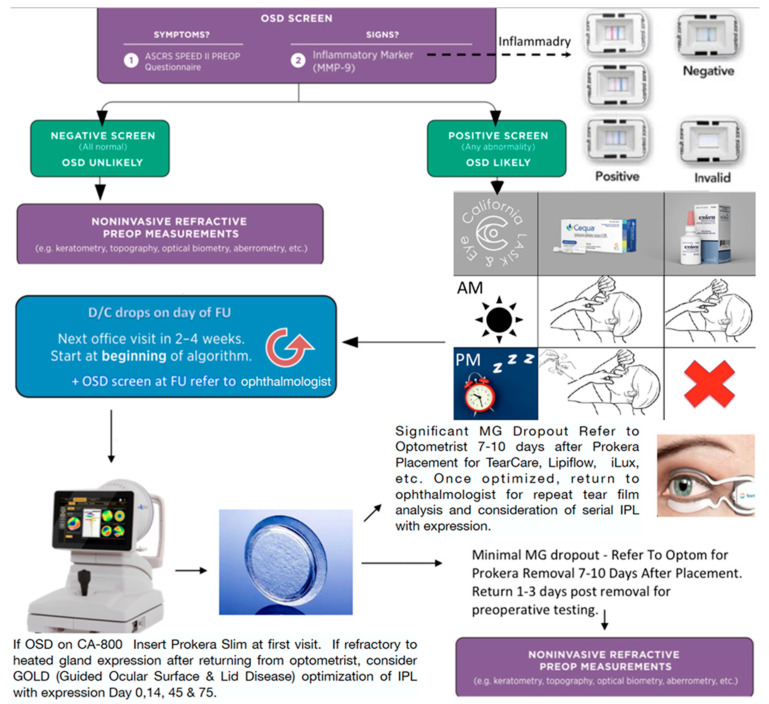
Guided Ocular Surface & Lid Disease (GOLD) Co-management algorithm. Incorporating aspects of the ASCRS algorithm one proposed GOLD co-management protocol involves first probing for signs with inflammatory for MMP-9 measurement (relatively low cost and placeable in every optometric office with a CLIA waiver). Symptoms can be established as well with SPEED II. Once uncovering ocular surface disease, the referring optometrist could first start the patient on a T-cell suppressant regimen such as BID cequa and QDAY Eysuvis. Patient would then first see the surgeon in 2–4 weeks after beginning the regimen. If tear film analysis revealed continued tear film instability and significant glandular disease, the surgeon would place a Prokera and refer back to the optometrist for meibomain gland optimization. Through interventions such as IPL and heated expression, the tear film can be stabilized as measured and help restore tear film stability. Once fully optimized, the stability of the tear film and meibomian gland health can be confirmed prior to Biometry and Topography to guide surgery.

**Figure 7 diagnostics-13-00837-f007:**
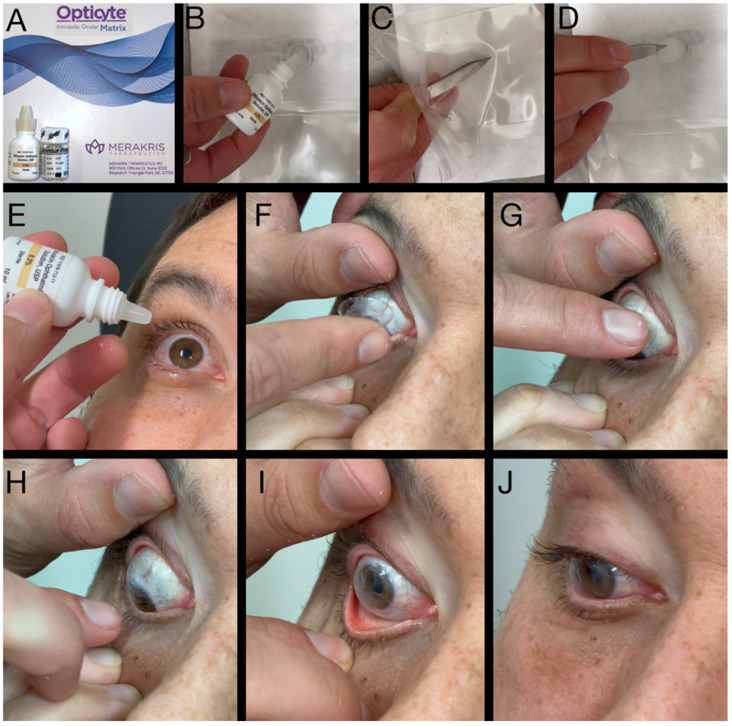
Speculum-Free Insertion of Amniotic Membrane Graft Under Kontur—Through use of a Kontur large diameter contact lens, an antibiotic drop, and an amniotic graft (**A**), a self-retaining amniotic membrane graft without a retaining ring can be created. A drop of antibiotic is placed inside the Kontur (**B**), and then the non-hydrated amniotic membrane is grasped with a pair of jeweler forceps (**C**). The amniotic membrane is placed with one edge touching the antibiotic fluid in the Kontur and the amniotic membrane sequentially adheres to the contact lens through hydrostatic attraction, similar to the application of a screen protector (**D**). Next an antibiotic drop is instilled in the eye (**E**) and then the patient is told to look down and pull down on their lower lid (**F**). The clinician then pulls up on the lower lid and places the Kontur with adhered AMG (**G**). Once on the eye (**H**) the patient is told to look down thereby centering the lens and AMG over the cornea (**I**). Next the patient and clinician release their fingers from the lids (**J**).

**Figure 8 diagnostics-13-00837-f008:**
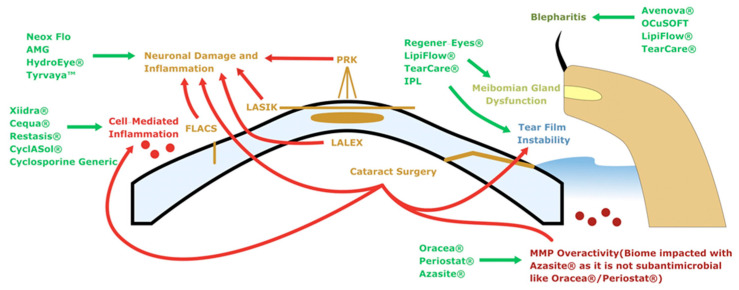
Solving STODS, a graphical summary. The ocular surface microenvironment undergoes unique perturbations from each surgical insult. A variety of therapeutics and interventions can be utilized to blunt the impact of STODS on the ocular surface.

**Table 1 diagnostics-13-00837-t001:** Summary of Unique Pathways to Surgical Temporary Ocular Discomfort Syndrome (STODS) Following Different Ocular Surgeries.

Ocular Surgery	Mechanism	Potential Therapeutic Targets
Cataract Surgery Along Cataract Surgery + MIGS	Ocular Surface Inflammation; corneal nerve loss; microbiome disruption	TYRVAYA™—Spares surface CEQUA^®^, RESTASIS^®^ HydroEye^®^ Regener-Eyes^®^
PRK LASIK	Corneal nerve loss, worst with LASIK	NEOX FLO AMG TYRVAYA™—Spares surface CEQUA^®^, RESTASIS^®^
LALEX	Corneal nerve loss, least of all laser-based refractive surgeries	Regener-Eyes^®^ TYRVAYA™—Spares surface CEQUA^®^, RESTASIS^®^
FLACS	Ocular Surface Inflammation, correlated with energy	TYRVAYA™—Spares surface CEQUA^®^, RESTASIS^®^ HydroEye^®^ Regener-Eyes^®^

**Table 2 diagnostics-13-00837-t002:** Strategies, Mechanisms, and Option of Various Treatments for Ocular Surface Optimization and Surgical Temporary Ocular Discomfort Syndrome (STODS).

Class	Mechanism	Examples	Duration	Special Uses/Comments
Lifestyle Changes	Preventing tear desiccation	Fan placement, avoiding prolonged reading	Indefinitely	This should become a permanent lifestyle change for all patients
Increasing Tear Films	Tear replacement; inflammatory marker dilution	REFRESH TEARS^®^ (Abbvie, Chicago, IL, USA)	QID, 2–4 months pre-op	Frequent use of unpreserved tears can worsen symptoms
	Punctal Occlusion	DuraPlug	2–4 months pre-op	Avoid using with some anti-inflammatories, diluting them in the tear film reduces effect
Hypchlorous Acid Scrubs Lotilaner Ophthalmic Solution	Meibomian exfoliation, mechanical debridement	Avenova^®^, OCuSOFT	2–4 months pre-op	For predominantly blepharitis or *Demodex* symptoms
Warming	Heat-based meibomian expression	Warm Compresses IPL	BID 2 months pre-op	Careful applying heat and debridement to inflamed tissue
	Heat-based meibomian expression	LipiFlow^®^, TearCare^®^	2 months pre-op	More regulated and gentle on eyelids
Pro-Biotics	Tear film support	HydroEye^®^	2 months pre-op	Surface sparing, easy to continue post-op
Anti-inflammatories	Corticosteroid	Prednisolone Acetate	BID 2 months pre-op	Monitor for pressure response
	LFA-1	Xiidra^®^	BID 2 months pre-op	Paired with Prednisolone Acetate BID
	Calcineurin inhibition	CEQUA^®^, RESTASIS^®^,	BID 2 months pre-op	Paired with Prednisolone Acetate QDAY
	Verenicline	TYRVAYA™	BID 1 month pre-op	Nasal spray; spares ocular surface; easy to continue post-op
	IPL	OptiLight	4 section pre-op	Can repeated post-op
MMP Inhibition	Doxycycline	ORACEA^®^, Periostat^®^	1–2 weeks pre-op	Minimal antibacterial properties so microbiome preservation
Advanced Tear Substitutes	Autologous Serum Tears	Serum Tears	2 months pre-op	Theoretically contain more patient-specific bio-compatible materials
Amnionic-Based Therapies	Amnionic Fluid-Based Tears	Regener-Eyes^®^	QID 2 months pre-op	Can be used under drug depot system (Hyper-CL™)
	Cryopreserved or Lyophilized Amnionic Membrane Grafts	AMG	Pre- or Post-Op	Can be placed under contact lens; consider for post-laser STODS
	Amnionic and Placental derived	NEOX FLO	Pre- or Post-Op	Can be placed under contact lens; consider for post-laser STODS

## Data Availability

Not applicable.

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
