# Peer review of "Solving STODS—Surgical Temporary Ocular Discomfort Syndrome"

_diagnostics, 2023, doi:10.3390/diagnostics13050837_

Round 1
Reviewer 1 Report
- This review makes a good contribution to the field of ophthalmology
- This review is written scientifically, structured, comprehensive, and easy to understand
- This review was written based on appropriate and adequate references
- The English and style used are correct, readable, and good (there are some typos that need to be fixed)
Author Response
The authors kindly thank the reviewer for the guidance to improve our manuscript. We have gone through the manuscript in attempt to resolve any further editing issues and have attached an edited final submission with track changes.
Kind Regards,
Brad Barnett
Reviewer 2 Report
In this review, the authors described Surgical Temporary Ocular Discomfort Syndrome (STODS). Guided optimization of Ocular Surface and Lid Disease (GOLD) is fundamental to successful refractive outcomes and mitigation of STODS. Understanding the molecular, cellular, and anatomic factors that affect the ocular surface microenvironment and the associated perturbations induced by surgical intervention can effectively optimize GOLD and prevent STODS. The authors attempted to outline a rationale for a tailored GOLD optimization depending on the ocular surgical insult. With the bench-to-bedside approach, authors will highlight clinical examples of effective GOLD perioperative optimization that mitigate STODS deleterious effect on preoperative imaging and postoperative healing. This review is compelling and interesting, the data is clear, and the discussions are adequate.
To make this review more profound, I would like to give three suggestions:
1. There is a spelling problem, such as "microenviroment" in the abstract, it should be "microenvironment".
2. There are two periods in paragraph 3 on page 2 and figure 7, it should be only one period.
3. Please discuss the influence of genetic factors on STODS.
Author Response
The authors appreciate the admonition to correct the spelling and grammar errors. We have made the requested edits.
We have also added a section on various genetic contributions to STODS. The authors had not considered discussing this but what a great suggestion. All changes highlighted in the final manuscript. Kind thanks for your assistance in improving our manuscript and considering it for your journal.
Best
Brad Barnett